# Alternative Pharmacological Strategies for the Treatment of Alzheimer’s Disease: Focus on Neuromodulator Function

**DOI:** 10.3390/biomedicines10123064

**Published:** 2022-11-28

**Authors:** Grace Cunliffe, Yi Tang Lim, Woori Chae, Sangyong Jung

**Affiliations:** 1Institute of Molecular and Cell Biology (IMCB), Agency for Science, Technology and Research (A*STAR), Singapore 138667, Singapore; 2Division of Neuroscience and Experimental Psychology, School of Biological Sciences, Faculty of Biology, Medicine and Health, University of Manchester, Manchester M13 9PL, UK; 3Faculty of Science, National University of Singapore, Singapore 117546, Singapore; 4Department of BioNano Technology, Gachon University, 1342 Seongnam-daero, Seongnam-si 13120, Republic of Korea; 5Department of Physiology, Yong Loo Lin School of Medicine, National University of Singapore, Singapore 117593, Singapore

**Keywords:** neuromodulation, neuropeptides, hormones, neurotrophins, ATP, metal ions, Alzheimer’s disease, synaptic plasticity, therapeutics

## Abstract

Alzheimer’s disease (AD) is a neurodegenerative disorder, comprising 70% of dementia diagnoses worldwide and affecting 1 in 9 people over the age of 65. However, the majority of its treatments, which predominantly target the cholinergic system, remain insufficient at reversing pathology and act simply to slow the inevitable progression of the disease. The most recent neurotransmitter-targeting drug for AD was approved in 2003, strongly suggesting that targeting neurotransmitter systems alone is unlikely to be sufficient, and that research into alternate treatment avenues is urgently required. Neuromodulators are substances released by neurons which influence neurotransmitter release and signal transmission across synapses. Neuromodulators including neuropeptides, hormones, neurotrophins, ATP and metal ions display altered function in AD, which underlies aberrant neuronal activity and pathology. However, research into how the manipulation of neuromodulators may be useful in the treatment of AD is relatively understudied. Combining neuromodulator targeting with more novel methods of drug delivery, such as the use of multi-targeted directed ligands, combinatorial drugs and encapsulated nanoparticle delivery systems, may help to overcome limitations of conventional treatments. These include difficulty crossing the blood-brain-barrier and the exertion of effects on a single target only. This review aims to highlight the ways in which neuromodulator functions are altered in AD and investigate how future therapies targeting such substances, which act upstream to classical neurotransmitter systems, may be of potential therapeutic benefit in the sustained search for more effective treatments.

## 1. Introduction

Alzheimer’s disease (AD) is a progressive neurodegenerative disorder which presents predominantly in the elderly. AD is the leading cause of dementia and is characterised by extensive brain atrophy, resulting in debilitating memory loss and executive dysfunction. It is estimated that approximately 50 million people are currently living with AD and other forms of dementia, and this is expected to quadruple in the next 30 years due to ageing of the global population [1]. By 2050, as many as 1 in 85 people are likely to be suffering from the disease [2], which is currently ranked as the fifth leading cause of death worldwide [3,4]. Despite such concerning figures, due to its complex and multifactorial nature, the exact causes of AD are yet to be elucidated, and current treatments remain insufficient at reversing pathology and act only to slow the inevitable progression of the disease and its associated symptoms. Additionally, patients with AD often show comorbidity with other diseases, making the development of treatment all the more challenging [5].

The majority of AD treatments target neurotransmitter systems; galantamine, rivastigmine, donepezil and tacrine are acetylcholinesterase (AChE) inhibitors, and thus enhance cholinergic transmission. Attenuated cholinergic transmission in brain regions including the basal forebrain, hippocampus, amygdala and cortex is well-known to occur as a result of AD presentation [6,7,8,9] and these findings form the basis of the cholinergic hypothesis, which is thought to be a leading contributor to AD progression [10,11]. Memantine is an antagonist of glutamatergic N-methyl-D-aspartate (NMDA) receptors and therefore reduces glutamatergic transmission, excessive levels of which have been shown to contribute towards excitotoxicity and neuronal death in AD [12]. Current treatments have unpleasant associated side effects; cholinesterase inhibitors often result in nausea, gastrointestinal upset and diarrhoea due to peripheral effects [13], and tacrine is rarely used in the modern day due to hepatotoxicity [14]. Additionally, major alterations to neurotransmission tend to present during advanced stages of disease progression, following the appearance of pathological proteins amyloid beta (Aβ) and tau [15], and pathologic cascades involving mitochondrial dysfunction and oxidative stress [16,17]. Consequently, drugs targeting aberrant neurotransmission, albeit better than nothing, do little to reverse progression or tackle the pathological roots of the disease.

The accumulation of Aβ plaques and tau neurofibrillary tangles has long been implicated as a major pathological hallmark of AD, and the aggregation of both proteins is viewed as a leading contributor to neuropathology, synaptic loss and associated cognitive decline [4,18,19,20]. Research relating to the development of drugs which aim to reduce the aggregation or increase the clearance of Aβ and tau has therefore been extensively undertaken, but such treatments have thus far exhibited notoriously high failure rates in clinical trials [21]. The recently FDA-approved drug aducanumab represents a step forward in terms of pathological targeting. As a monoclonal antibody against Aβ, the drug is the first to target pathology upstream of neurotransmission, and has been suggested to reduce Aβ plaque burden and slow cognitive decline in patients with prodromal or mild AD [22]. However, significant controversy surrounds its accelerated approval due to observations of harmful side effects and limited efficacy [23]. As a result, the drug was rejected for use by the European Medicines Agency, and remains yet to be approved outside of the United States [24]. The causes of AD have not been fully elucidated, but they are likely a combination of genetic and environmental factors which result in Aβ and tau protein aggregation, disrupted neuronal function, synaptic plasticity deficits, neuroinflammation, oxidative stress and neurodegeneration which underlie cognitive dysfunction. A major limitation of current AD treatments is that they target just one component of this extremely complex, multifactorial disorder, and efficacy is further compromised due to the low ability of conventional drugs to cross the blood-brain barrier (BBB) [25]. There is therefore an urgent need to expand the focus of AD targets and delivery methods.

Neuromodulators are substances released by neurons which influence neurotransmitter release and signal transmission across synapses. These include, but are not limited to; neuropeptides, hormones, neurotrophins, adenosine triphosphate (ATP) and metal ions. Neuromodulators play a vital role in controlling neuronal function and communication, and many have been observed to undergo alterations in AD, resulting in a loss of normal physiological function and hence aberrant neuronal activity which underlies associated cognitive deficits. Changes to neuromodulator levels impact key downstream signalling pathways and the activation of neurobiochemical mediators, which contributes to increased neurodegeneration, atypical synaptic plasticity and the accumulation of pathological proteins. Additionally, divergent neuromodulator function compromises neuronal and metabolic homeostasis within the brain, resulting in oxidative stress and excessive activation of the immune response, further advancing disease progression. Neuromodulator signalling therefore impacts widespread and numerous components of AD progression, making it a promising therapeutic target. This review will discuss the roles of neuromodulators in neuronal homeostasis and synaptic plasticity, and how their function is modified over the course of AD, contributing to cognitive deficits. The ways in which neuromodulator physiology may be targeted in order to better treat multiple components of AD upstream of abnormal neurotransmission will be considered, as well as the potential of neuromodulator nanoparticle drug delivery in overcoming additional limitations of conventional drugs, such as associated side effects and BBB impermeability.

## 2. Neuromodulator Function in the Central Nervous System and Dysregulation in AD

In order to determine how alterations to neuromodulator function may underlie aberrant neuronal activity which presents as a result of AD pathology, the processes by which neuromodulator release occurs and how neuromodulators exert effects on a range of targets in the central nervous system (CNS) must first be understood. Many studies have investigated how the activity of neuromodulators, as opposed to neurotransmitters alone, underlie essential processes involved in neuronal function, metabolic homeostasis and synaptic plasticity, and how these display alterations over the course of AD. A summary of neuromodulator dysfunction in AD, and how this contributes to the progression of the disease, is shown in Figure 1.

### 2.1. Neuropeptides

Neuropeptides are small, endogenous messenger molecules which co-exist in neurons alongside classical transmitters. Upon release from neurons, they act locally or over long distances, and are able to exert a vast array of very fast (milliseconds) or long-lasting (seconds to minutes) effects on autocrine and paracrine targets [88]. Interactions between neuropeptides and neurotransmitters, alongside their independent activities, are vital for the modulation of neuronal activity associated with neurodevelopment, neuronal homeostasis, sensory perception, the inflammatory response and metabolic function [89]. Critically, neuropeptide binding to G-protein-coupled receptors (GPCRs) in the hippocampus is able to activate numerous signalling pathways in order to modulate synaptic plasticity. For example, orexin has been shown to couple to Gq protein transducers, which results in the activation of phospholipase C. This leads to cytosolic calcium influx and the downstream activation of protein kinase C and the extracellular signal-regulated kinase (ERK) 1/2 signalling pathway [90]. Activation of this pathway has been strongly implicated in the formation and retrieval of long-term memory [91]. Oxytocin and insulin have been reported to act through the phosphatidylinositol 3-kinase (PI3K)/protein kinase B (AKT) signalling pathway in order to modulate excitatory synaptic transmission, long-term potentiation (LTP) and long-term depression (LTD) in the hippocampus [92,93,94]. Vasoactive intestinal polypeptide (VIP) and pituitary adenylate cyclase-activating polypeptide (PACAP) have been linked with synaptic plasticity through activation of the Gα protein family, which stimulates cyclic adenosine monophosphate (cAMP) and protein kinase A (PKA) signalling [95,96]; PKA has been reported to be required for hippocampal-dependent memory formation [97]. Neuropeptide Y (NPY) activates the c-Jun NH2-terminal kinase (JNK) signalling pathway, a form of mitogen-activated protein kinase (MAPK) signalling which leads to LTP attenuation and the control of neuronal excitability [98]. NPY additionally contributes towards the control of neuronal excitatory [99] and inhibitory [100] transmission, in order to maintain the balance of excitation and inhibition in neural networks. Corticotropin-releasing factor (CRF) has been shown to have opposing effects to neuropeptide Y in the control of inhibitory transmission [100], whilst substance P facilitates excitatory glutamatergic transmission by prolonging activation of NMDA receptors [101]. Neuropeptides including neuropeptide Y, calcitonin gene-related peptide (CGRP), PACAP and bradykinin have also been shown to exert neuroprotective effects by reducing excitotoxicity and apoptosis and exerting anti-inflammatory and antioxidant actions on neurons following brain injury [102,103]. Therefore, neuropeptides have essential roles in the control of neuronal homeostasis and recovery following insult to the brain, excitatory/inhibitory neurotransmission, and synaptic plasticity associated with learning and memory.

### 2.2. Neuropeptides; Dysfunction in AD

Neuropeptide levels have been observed to undergo widespread changes in AD patients and animal models, and the extent of these changes has been suggested to correlate with disease severity [104]. Alterations to neuropeptide levels occur as a result of AD pathology, but are critically also a contributing factor towards its progression. NPY levels are reduced in AD patients and mouse models [26,27,28], causing a decline in the neuroprotective effects exerted by the neuropeptide, including its ability to attenuate the toxic accumulation of Aβ [28], prevent calcium-induced accumulation and excitotoxicity in presynaptic terminals [105], regulate immune cell function [106] and induce neurogenesis in areas which undergo large neuronal loss, such as the hippocampus [28,107]. Altered NPY levels are also likely to influence the control of synaptic plasticity and excitatory and inhibitory transmission, due to the reported roles of the neuropeptide in these processes, leading to deficits in memory formation [98,99,100]. Notably, the number of NPY-expressing interneurons in the hippocampus was shown to be reduced in AD mice as young as 1-month of age, long before the appearance of cognitive symptoms or Aβ accumulation [108], suggesting that alterations to NPY activity occur upstream of these processes.

The levels of neuropeptides including oxytocin [29], insulin [30], VIP [31] and PACAP [32], all of which contribute to the control of synaptic plasticity in the hippocampus, have displayed adjustments in AD, resulting in aberrant control of LTP and LTD and subsequent interference of memory formation. Substance P [33] and CRF [34], which contribute towards regulation of the excitatory/inhibitory balance of neural networks, also display altered levels over the course of the disease. Reductions in CRF immunoreactivity in the cerebral cortex have been correlated with lower levels of choline acetyltransferase (ChAT) activity, which impacts the biosynthesis of acetylcholine and hence may be an important contributor to the cholinergic loss observed in AD [35]. Levels of opioid peptides such as dynorphin, hemorphin and enkephalin have been reported to increase in AD patients [36,37,38] and mouse models [39,40], correlating with increased Aβ plaque burden and altered glutamatergic receptor expression underlying learning and memory deficits [109,110]. The role of opioid neuropeptides in AD pathology appears receptor dependent; a recent study has suggested that activation of δ- and µ-receptors has opposing effects on the β-site APP cleaving enzyme 1 (BACE1) [111]. Therefore, activation of opioid receptors is likely to have differential effects on APP cleavage and the production of Aβ. Notably, alterations to opioid levels have been reported to disrupt activity of a number of other neurotransmitters, including gamma-aminobutyric acid (GABA), acetylcholine, serotonin and noradrenaline [109,112], reaffirming that changes to neuropeptide levels likely occur upstream of aberrant neurotransmitter activity.

Finally, altered neuropeptide levels have been heavily associated with cognitive disturbances in AD in addition to those classically associated with memory decline and executive dysfunction. Oligomeric Aβ has been suggested to modulate voltage-gated calcium channel activity on NPY-expressing neurons in the arcuate nucleus of the hypothalamus, which may underlie changes to body weight regulation during early AD stages [113]. The overexpression of orexinergic signalling in the hypothalamus has been linked with sleep disturbances, and has been implicated as a contributor to increased accumulation of Aβ and tau pathology [41,42,43,44]. Neuropeptide changes therefore contribute towards numerous components of AD, with many occurring upstream of neurotransmitter dysfunction and pathological Aβ and tau presentation.

### 2.3. Hormones

Hormones are chemical messengers which are present in and act upon both CNS targets and peripheral tissues and organs. A number of classical hormone release pathways have been well-characterised, such as the hypothalamic-pituitary-adrenal (HPA) axis involved in cortisol release underlying the body’s response to stress [114], the functions of oestrogen, progesterone, luteinizing hormone (LH) and follicle-stimulating hormone (FSH) in the menstrual cycle [115], and testosterone in male development [116]. In addition to classical roles in homeostatic processes, hormones have been reported to impact cognitive functioning of the brain [117]. For example, glucocorticoids (GCs) such as cortisol have been associated with the neuromodulation of memory related with stress and fear via actions on neurons in limbic regions [118,119]. Cortisol binding to glucocorticoid receptors in the brain is tightly regulated and finely balanced. Hippocampal neurons are particularly sensitive to the hormone; moderate release is known to enhance memory consolidation, but hyposecretion and hypersecretion have both been associated with memory impairments [117].

Despite its more common association with functions in the reproductive system, the release of oestrogen has also been suggested to exert neuroprotective effects on neuronal activity and underlie memory processes linked to spatial and object recognition, as well as fear memory, via actions on hippocampal neurons which express the oestrogen receptor [120]. Oestrogen has been reported to modulate synaptogenesis, dendritic branching and myelination of neurons involved in the release of neurotransmitters in serotonergic, dopaminergic, GABAergic and glutamatergic systems, directly affecting neuron structure, function and signalling [121]. In addition to its roles in the female reproductive system and cognitive function, oestrogen is produced in males via the conversion of testosterone (T) by the aromatase enzyme [122]. Testosterone modulates neuronal function and gene expression underlying behaviours associated with aggression, social interaction and anxiety via actions on androgen receptors (ARs) in brain regions including the amygdala, hypothalamus and thalamus [123]. Additionally, testosterone depletion has been associated with hippocampal dysfunction and associated learning and memory deficits [124], as the androgen has been shown to regulate the expression of microtubule-associated protein 2 (MAP2) and postsynaptic density protein 95 (PSD95) [125], cytoskeletal and postsynaptic proteins respectively, which are essential for synaptic integrity and plasticity.

Additional sex hormones including LH and FSH have been implicated in cognitive function; receptors for the hormones are found in hippocampal regions critical for learning and memory [126,127]. LH has been suggested to regulate signalling pathways associated with the expression of glycogen synthase kinase 3β (GSK3β), phosphorylated calcium/calmodulin-dependent protein kinase II (CAMKII) and phosphorylated α-amino-3-hydroxy-5-methyl-4-isoxazole propionic acid (AMPA) receptor subunit glutamate receptor 1 (GluR1) [128,129], all of which are implicated in synaptic plasticity underlying memory processes in the hippocampus. Less is known about the impact of FSH signalling in the brain, although FSH receptors have been found in the hippocampus [126] and the hormone has been linked with cognitive functioning in postmenopausal women [130]. Finally, progesterone has been shown to modulate neuronal growth and excitatory transmission, by increasing dendritic spine density [131] and the expression of AMPA receptors both directly [132] and indirectly through the upregulation of brain-derived neurotrophic factor (BDNF) signalling in hippocampal synapses [133]. The actions of sex hormones in prefrontal and limbic areas are therefore likely to contribute towards cognitive processes associated with memory and executive function.

### 2.4. Hormones; Dysfunction in AD

Glucocorticoid levels are well-known to display significant increases in both the periphery and brain of AD patients [134,135,136]. Recent studies have proposed that these changes do not simply arise as a result of AD progression, but may in fact be critical drivers of disease pathology. The hypersecretion of glucocorticoids has been linked with increased aggregation of Aβ in rodent models [79]. This may be due to the presence of a glucocorticoid responsive element (GRE) in the promoter region of the β-secretase gene, glucocorticoid binding to which augments expression of β-secretase and subsequent amyloidogenic processing [80,137]. Increased glucocorticoid levels have also been proposed to contribute towards excessive tau hyperphosphorylation through upregulation of GSK3β activity [81]. Excessive glucocorticoid receptor activation associated with early life stress has been suggested to contribute towards synaptotoxicity and increased microglial-mediated inflammation, resulting in reduced hippocampal LTP and disrupted memory retrieval [82,138,139]. Exposure to chronic stress at postnatal days 2–9 was sufficient to significantly increase cortisol levels and Aβ accumulation, and hamper reversal learning once APP/PS1 mice reached 12 months of age [140], highlighting the early influence of cortisol level alterations on AD presentation. Excessive circulation of glucocorticoids impacts numerous components of pathology associated with hippocampal atrophy, and it is therefore being increasingly considered as an important initiator of AD [117,141,142].

One major area of interest with regard to the modulation of hormone levels in AD is that of oestrogen levels in female patients. It is commonly known that more women present with AD than men; around two thirds of cases occur in females [143]. Although this may be due to sex differences outside of differential sex hormone circulation (such as societal gender elements or other sex-specific differences in cell function [144]), this could also be attributed to differences in sex hormones and their function in the brain, based on hormone therapy studies and the proven roles of sex hormones in cognition. Studies have suggested that a reduction in oestrogen during the menopause may alter regulation of the APOE gene coding for the apolipoprotein E [145], a highly recognised risk factor gene for familial AD. This results in changes to lipid transport, glucose homeostasis, mitochondrial function and neuroinflammatory processes, and hence impacts neuronal activity as well as increasing Aβ aggregation [73]. A rodent study has shown that oestrogen knock-out in APP/PS1 mice leads to the exacerbated formation and attenuated clearance of Aβ plaques [146], further emphasising its neuroprotective functions. AD presents most commonly in older women, and consequently the increased release of LH and FSH during and following the menopause and their potential roles in disease development has also been considerably researched. In contrast to that observed with oestrogen, studies have reported significantly elevated serum levels of both gonadotropins in AD patients [83], which has been strongly associated with a decline in hippocampal-dependent cognitive function [84,85,86]. This may be attributed to a gonadotropin-mediated increase in pathological protein appearance; FSH has been shown to increase the expression of the CCAAT-enhancer-binding protein β (C/EBPβ), a transcription factor which activates a δ-secretase known to cleave APP and tau to form pathological Aβ and aggregated tau [86]. LH activates a number of second messenger signalling cascades involving protein kinases A and C and MAPKs, which contribute to increased amyloidogenic APP processing and tau hyperphosphorylation [87]. Changes to levels of oestrogen, LH and FSH as a result of the menopause can thus be viewed as potential underlying contributors to the development of AD via the increased production of pathological Aβ and tau.

Despite a more prominent focus on female hormone dysregulation in AD, elderly men with lower serum levels of testosterone also present with significantly higher risk of cognitive dysfunction and AD development [147]. The activation of androgen receptors by testosterone leads to increased expression of α-secretase and reduced production of β-secretase, stimulating non-amyloidogenic APP processing over amyloidogenic processing [148,149]. Therefore, T directly impacts the production and clearance of Aβ through androgen receptor signalling pathways. Testosterone has also been associated with the regulation of tau phosphorylation as it controls the activation of signalling pathways involving protein kinase B and glycogen synthase kinase 3β [150]. The hormone exerts protective effects on neuronal synapses by preserving the production of presynaptic proteins including synaptophysin, synaptotagmin and synapsin-1 [74,75], and has thus been reported to contribute towards the maintenance of dendritic spine density, synaptic integrity and synaptic plasticity in the hippocampus, which underlie learning and memory propensity [76,77,78]. Depletion of testosterone in the hippocampus has been shown to induce oxidative stress and neuronal damage, leading to a reduction in antioxidants, upregulated expression of caspase-3 and subsequent neuronal apoptosis [151]. Reduced levels of the hormone have consequently been associated with Aβ accumulation, tau hyperphosphorylation, oxidative stress and neuron loss which contribute to AD progression [74,78,147,152,153].

### 2.5. Neurotrophins

Neurotrophins are growth factor proteins which regulate the development, survival and function of neurons and glia. Mammalian neurotrophins consist of brain-derived neurotrophic factor (BDNF), nerve-growth factor (NGF), neurotrophin-3 (NT-3) and neurotrophin-4 (NT-4). Each neurotrophin exerts effects via binding to specific receptors, promoting cell signalling pathways associated with various neuronal processes, including differentiation, maturation and proliferation [154]. One essential role of neurotrophins is to regulate the balance of neuronal survival and apoptosis. The promotion of apoptosis occurs following activation of the p75 neurotrophin receptor (p75NTR), for which all neurotrophins have similar binding affinity [155]. Conversely, cell survival is initiated by neurotrophin binding to the cell survival-promoting tropomyosin-related kinase receptors (TrK-Rs) [154]. Activation of these two receptor types by neurotrophins therefore plays a critical role in the determination of neuronal survival.

The differential activation of p75NTR and TrK-Rs also has contrasting influences on neuronal synaptic activity; p75NTR activation has been observed to result in LTD in the hippocampus via altered AMPA receptor expression [156]. TrkB receptor activation by BDNF, on the other hand, promotes LTP of hippocampal neuronal synapses which is dependent on the recruitment of phospholipase C [157], and increased release of BDNF has been shown to occur during synaptic remodelling [158,159]. Therefore, the balance of neurotrophin activity at receptors with opposing effects on synaptic plasticity acts to modulate synaptic strength associated with learning and memory processes [160]. Aside from phospholipase C, other major downstream intracellular signalling cascades associated with BDNF binding to the TrkB receptor include PI3K/GSK3β/Akt and ERK/MAPK pathways [154,161,162]. Activation of these signalling pathways leads to the upregulation of specific transcription factors and second messengers, which control gene transcription associated with ion channel and neurotransmitter expression underlying synaptic function [154,162].

Although BDNF has emerged as the key molecule underlying neurotrophic control of synaptic plasticity in the hippocampus, neurotrophin-4 is also recognised to bind to and activate the TrkB receptor [163]. Application of NT-4 to cultured rat hippocampal neurons enhanced AMPA-R-mediated glutamatergic synaptic transmission similarly to that observed following BDNF addition, suggesting a corresponding role in plasticity [164], although the exact role of NT-4 in LTP induction has been relatively understudied. NGF infusion into the rat brain led to increased septal cholinergic innervation of the hippocampus, and the facilitation of hippocampal LTP induction. Conversely, NGF blockade significantly reduced LTP induction and led to impaired spatial memory, suggesting that, via actions on TrkA receptors, NGF is essential for cholinergic-dependent hippocampal synaptic plasticity [165]. Neurotrophin-3 binds to the TrkC receptor, and has been linked with the facilitation of synaptic function, enhanced synaptic strength and inhibition of GABAergic transmission in both developing and mature neurons [166,167,168,169]. Therefore, despite a prominent focus on BDNF in more recent years, all four mammalian neurotrophins have been proposed as modulators of synaptic plasticity.

### 2.6. Neurotrophins; Dysfunction in AD

Due to their roles in the modulation of synaptic strength underlying LTP and learning processes in the hippocampus, it is unsurprising that the modulation of neurotrophin activity, particularly that of BDNF, is altered in AD. BDNF mRNA is reduced in the hippocampus of AD patients [60], correlating with attenuated BDNF serum levels in a disease severity-dependent manner [61]. BDNF levels have also been shown to be significantly decreased in hippocampal extracts from AD mouse models [62,63]. Additionally, knock out of the TrkB-R in 5xFAD mice leads to the exacerbation of memory-related behavioural impairments, reduced hippocampal expression of AMPA and NMDA receptor subunits and alterations to pathways downstream of BDNF signalling involving ERK-2 and GSK3β activation [170,171]. ERK-2 and GSK-3β are regulators of tau phosphorylation, and their aberrant production and function have been linked to hyperphosphorylation and the resulting production of tau neurofibrillary tangles, which contribute to cognitive dysfunction [69,70,170]. Increased GSK3β activity has also been linked with the promotion of amyloidogenic processing of APP, leading to elevated deposition of Aβ plaques [71], whilst sustained activation of ERK-2 leads to neuronal death [172], resulting in elevated neurodegeneration and synaptic dysfunction. Aβ oligomers were recently shown to bind p75NTRs in the nanomolar range, leading to the exacerbation of Aβ-mediated dendritic spine loss in hippocampal neurons and resulting synaptic dysfunction and neuron loss [72]. Alterations to neurotrophin-mediated Trk-R and p75NTR signalling have therefore been implicated as key mediators in the progression of early pathological hallmarks of the disease.

The roles of NT-4, NT-3 and NGF in AD are less elucidated; a handful of studies have observed no significant alteration to NT-3 levels in AD patients [60,173,174], although significant reductions in NT-3 levels in the motor cortex have been reported [64]. In the same study, elevated NGF levels were noted in the dentate gyrus, although no significant change in NGF levels across all brain regions has also been observed [60]. A separate report revealed reduced NGF levels, correlating with decreased ChAT activity, at the onset of Aβ plaque presentation, and increased NGF levels during more advanced disease stages [65]. TrKA-R expression, required for NGF signalling, was observed to be significantly reduced in the parietal cortex of AD patients [66], and the neurotrophin has been suggested to undergo degradation during both pre-clinical and clinical stages of AD [67]. Levels of neurotrophin-4 have been reported to display a small reduction in the hippocampus and cerebellum of AD patients [68], although aside from this report very few studies have investigated changes to NT-4 levels and function in AD. Considering their reported roles in hippocampal synaptic plasticity, the lack of research into neurotrophins aside from BDNF in an AD context is surprising. Further research into whether and how NGF, NT-3 and NT-4 display alterations in AD patients and mouse models is likely to be worthwhile.

### 2.7. ATP

Adenosine 5′ triphosphate (ATP) is widely recognised for its role as the main source of energy to cells in both the brain and periphery, which is critical in enabling cell respiration, differentiation and function. In addition to its classical role in energy supply, ATP is essential for maintaining ionic concentrations across neuronal membranes. This is heavily dependent on the activity of P-ATPases, which harness ATP to pump ions across the neuronal membrane. The membrane potential is governed by the concentration of Na^+^ and K^+^ ions either side of the neuronal membrane, which is tightly controlled by Na/K-ATPase [175,176]. Na/K-ATPase maintains low intracellular Na^+^ and high intracellular K^+^ by actively transporting Na^+^ out and K^+^ in to the neuron [177]. In parallel, the P-ATPase Cl–ATPase sustains a low Cl^−^ ion concentration intracellularly [178], thus exerting inhibitory control of neuronal excitability and enabling protection against excitotoxic stimuli [56]. The availability of ATP and subsequent actions of P-ATPases therefore directly impact neuronal excitability and protection against excitotoxicity.

ATP has been reported to act on two purinergic receptors; metabotropic P2Y and ionotropic P2X receptors [179,180,181]. Binding to P2X receptors leads to the opening of ligand-gated ion channels permeable to Na^+^, Ca^2+^ and K^+^, hence enabling the control of neuronal excitability and cell signalling [182,183,184]. Actions of ATP released from both neurons and glia on these receptors have also been associated with postsynaptic AMPA receptor trafficking at glutamatergic synapses through PI3K activation, enabling the control of slow neuromodulatory activity which influences synaptic strength [185,186,187]. ATP release from astrocytes has been suggested to regulate NMDA receptor and PSD95 expression associated with LTP induction, implicating ATP as a critical modulator of neuron-glia communication underlying synaptic plasticity [188]. ATP binding to G-protein-coupled P2Y receptors has also been shown to modulate synaptic plasticity, neurotransmitter release and ion channel activity in brain regions including the prefrontal cortex and hippocampus [189].

### 2.8. ATP; Dysfunction in AD

The brain is extremely vulnerable to metabolic alterations, and disruption to the production of ATP impacts neuronal function in a multitude of ways. The accumulation of tau neurofibrillary tangles has been reported to contribute to alterations in mitochondrial transport, dynamics and bioenergetics [190] and subsequent ATP production, whilst Aβ has been shown to incorporate with mitochondria, particularly in hippocampal regions with high synapse density and associated energy demand. This results in the presentation of mitochondria with abnormal morphology [191,192]. Dysfunction in the production of ATP as a result of mitochondrial damage and a metabolic switch from oxidative phosphorylation to glycolysis, reminiscent of the Warburg effect observed in cancer cells, has recently been proposed to act as a driver of neurodegeneration in AD. Traxler et al., [53] found that increased production of pyruvate kinase (PKM) 2 (an essential glycolytic enzyme, which catalyses the conversion of phosphoenolpyruvate to pyruvate) as opposed to PKM1 in patient-derived induced neurons may be responsible for initiating this switch, correlating with an increase in the PKM2 to PKM1 ratio in the prefrontal cortex of AD patients. Evidence for increased glycolytic flux in AD has been reported in further studies, which have observed early alterations in glucose metabolism in rodent models of AD [193] and significant increases in glycolytic-associated enzymes including PKM2 and lactate dehydrogenase in frontal and temporal brain regions of AD patients [194]. A reduction in ATP levels has been associated with the dysfunction or reduced levels of F_1_F_O_-ATP synthase [195] and cytochrome c oxidase [196], critical enzymes involved in the final step of oxidative phosphorylation within the mitochondrial membrane. A switch from oxidative phosphorylation to glycolysis has been shown to occur in microglia as well as neurons; microglia collected from the brains of human AD patients and 5xFAD mice presented with higher lactate and PKM2 levels and increased transcription of glycolytic genes. These changes correlated with the upregulation of the microglial inflammatory response and the release of pro-inflammatory cytokines such as interleukin 6 (IL-6) and tumour necrosis factor α (TNF-α) [54]. Additionally, PKM2 has previously been reported to upregulate γ-secretase activation, leading to the enhancement of Aβ production in the 3xTg-AD mouse model [197].

Reduced generation of ATP as a result of metabolic switching has a direct and profound impact on neuronal activity, as sufficient ATP levels are critical for regular functioning of P-ATPases. Activity of Na/K-ATPase and Cl-ATPase was shown to be reduced in the brains of AD patients, leading to reduced gradients of Na^+^, K^+^ and Cl^−^ ions and subsequent excitotoxicity and neuronal death [56]. The inhibition of Na/K-ATPase as a result of Aβ binding to the enzyme has been suggested to exacerbate cell excitotoxicity by leading to increased calcium influx into neurons [198], and the intensification of ionic imbalances. Additionally, soluble Aβ oligomers are able to form pores in the neuronal membrane, resulting in leakage of intracellular ATP into the extracellular space [55]. Excessive extracellular ATP binding to P2X receptors leads to additional calcium influx into neurons, further exacerbating excitotoxicity and cell death [57,58]. A role for aberrant ATP activity in AD presentation is further evidenced by the observation that Aβ oligomer application results in overexpression of P2X receptors in cellular models of AD, which precipitates the disruption of synaptic transmission at both presynaptic and postsynaptic neuronal regions [57]. An overabundance of extracellular ATP also exacerbates the inflammatory response and increases pro-inflammatory cytokine release, due to the expression of P2X and P2Y receptors on, and subsequent hyperactivity of, glial cells [199,200]. Therefore, the overactivation of purinergic receptors may be a critical mediator of Aβ-induced toxicity.

It is apparent that pathological proteins contribute to compromised function of mitochondria and ATP generation, but alterations to brain metabolic activity in AD have been suggested to precede their accumulation, and act as a critical driver of disease progression [192,201]. Reduced mitochondrial respiration levels and increased oxidative stress were shown to occur as early as three months in 3xTg-AD mice, prior to the appearance of Aβ plaques or tau tangles [202]. Alterations to mitophagy, the process whereby abnormal mitochondria are degraded in lysosomes in order to ensure optimal quality control of mitochondrial function, has been proposed as a central contributor to metabolic dysfunction-associated disease progression [59]. Abnormal mitophagy has been suggested to occur during early stages of AD as a result of mitochondrial dysfunction and altered ATP production, leading to compromised clearance of pathological Aβ and tau and hence their increased accumulation [203]. Mitochondrial dysfunction and disruptions to ATP generation as a result of a metabolic switch reminiscent of the Warburg effect can therefore be seen as a key driver of AD disease progression, by increasing Aβ and tau accumulation, neuroinflammation, neuronal excitotoxicity and synaptic loss.

### 2.9. Metal Ions

The maintenance of a homeostatic balance of metal ions such as copper, zinc and iron is crucial for normal physiological functioning of the human brain. Copper is the third most abundant trace metal ion found in the brain [204], and is present in the highest concentrations within synaptic vesicles, indicating its involvement in neurotransmitter release and reuptake in the synapse [205]. The acute application of copper is found to have blocking effects on GABA- and NMDA-mediated neurotransmission [206,207,208]. Research in rat brain models has shown that copper has an inhibitory effect on LTP induction [209], which could be due to its ability to bind antagonistically to NMDA receptors. However, copper has also been suggested to inhibit LTP in the CA3 region of the mouse hippocampus via an NMDA-independent mechanism [210]. Upon prolonged exposure of primary hippocampal neurons to copper, AMPAergic neurotransmission is promoted, corresponding to accumulation of the PSD95 protein and concurrent clustering of AMPA receptors on the plasma membrane [211]. This increase in AMPAergic neurotransmission is found to be transient, indicating that copper’s neuromodulatory effects are under homeostatic regulation. Additionally, copper serves an important physiological role as a driver of redox reactions performed by enzymes. Under normal physiological conditions, copper serves as the redox-active metal in the metabolic production of energy via the electron transport chain, in the regulation of neurotransmitters, neuropeptides, and dietary amines [204,212].

Zinc is one of the most abundant trace metals found in the brain, second only to iron. It is highly involved in the structural and catalytic properties of more than 300 proteins, including those implicated in the protection of the brain from oxidative imbalance and stress following the sequestration of reactive oxygen species (ROS) [213,214,215]. Zinc is predominantly found in regions of the brain associated with higher cognitive functions [216] and within vesicles of presynaptic terminals of glutamatergic neurons [217]. The accumulation of zinc in presynaptic terminals is facilitated by the zinc transporter ZnT3 [218] and, upon neuronal activation, transiently increasing concentrations of zinc are observed in the synaptic cleft [219]. Synaptic release of zinc in response to neuronal activation has been found to modulate the activities of glutamatergic NMDA and AMPA receptors and ionotropic glycine receptors [220]. The metal has been reported to have inhibitory effects on GABA_A_ and NMDA receptors, while it has been observed to act on glycine receptors in a biphasic manner; exerting excitatory effects at concentrations lower than the micromolar range, and inhibitory effects at concentrations lower than the millimolar range [216]. As a result of its impacts on neurotransmission, zinc has been suggested to act as a modulator of synaptic plasticity, LTP and LTD [221,222,223].

Iron is the trace metal present in the highest concentration in the brain. The maintenance of iron concentration is very tightly controlled, as it is crucial in enabling normal physiological function while preventing harmful oxidative effects. Iron is an important cofactor for various enzymes, for example, those responsible for ATP production, myelination and synthesis of DNA, RNA, proteins and neurotransmitters [224]. The brain is the only organ in the body that requires a constant supply of readily available iron, and deficiency leads to neurological and cognitive dysfunction. Due to its role as a redox cycling metal, iron is a critical contributor to mitochondrial function, neuronal myelination and neurotransmitter anabolism and catabolism [225]. It has also been suggested to play a role in synaptic plasticity via the generation of ryanodine receptor (RyR)-mediated calcium release following NMDA-R stimulation and subsequent activation of ERK1/2 signalling, which is essential for sustained LTP in the hippocampus [226].

### 2.10. Metal Ions; Dysfunction in AD

Trace metals have a number of reported roles in neurotransmission underlying synaptic plasticity and neuronal activity, as well as the protection of the brain against oxidative stress, and their abnormal accumulation has thus been linked with the progression of AD pathology and synaptic deficits. According to post-mortem analysis of AD patients, there is excessive accumulation of iron, zinc and copper colocalising with amyloid beta plaque deposition [45]. The ability of Aβ to initiate oxidative stress has been proposed to depend on its interactions with and reduction of redox metal ions, which results in the production of ROS such as hydrogen peroxide [49].

Research regarding the role of copper ions in AD pathogenesis is ambiguous [227]. A number of studies report a drastic reduction in copper levels in the brains of AD patients, with content negatively correlating with degree of severity of amyloid plaque presentation [228,229]. The absence of copper has been demonstrated to result in the amyloidogenic cleavage of APP molecules leading to the production of Aβ. Further evidence shows that copper ions are able to block the production of Aβ via interaction with a γ-secretase complex [230], or by interfering with the dimerization of APP [231]. Conversely, many reports implicate increases in copper levels and their association with Aβ as a major contributor to accelerated plaque aggregation, production of reactive oxygen species and associated disease pathology [46,47,50,232,233]. These reported inconsistencies may be due to differential roles of oxidized and reduced copper ions in the formation of amyloid fibrils [234]. Copper binding to Aβ has also been associated with the compromised control of excitatory and inhibitory transmission, resulting in synaptotoxicity and learning and memory dysfunction [235,236].

Zinc was first implicated as a possible risk factor for dementia in 1981 [237]. Since then, evidence of high concentrations of zinc incorporation with Aβ has been reported, contributing to plaque formation under physiological conditions [45,238,239]. Studies in AD patients have postulated a redistribution of zinc in the brain, and the number of zinc transporters has shown an increase in expression level during early stage AD, prior to the appearance of clinical manifestations [48]. The binding of zinc to APP at Lys 16 results in its inability to be cleaved by alpha-secretase, and thus results in the increased production of amyloid beta [48]. In addition, zinc has been discovered to promote MAPK-dependent signaling pathways, which contribute to upregulation of tau phosphorylation and the subsequent appearance of neurofibrillary tangles [240,241,242]. Altered zinc homeostasis has been suggested to alter the expression of the presynaptic synaptosomal-associated protein of 25 kDa (SNAP-25), as well as PSD95, AMPA-Rs and NMDA-Rs, and also reduce neuronal spine density, resulting in synaptic deficits which underlie memory loss [243]. Finally, zinc dyshomeostasis has been reported to impair mitochondrial function, and lead to the production of toxic ROS and oxidative stress-induced cell death [51].

Iron levels in the frontal cortex of human AD patients have been observed to correlate with the severity of Aβ and tau deposition [244]. Similarly to copper and zinc, excessive iron accumulation has been reported to colocalize with Aβ, lead to increased aggregation of the pathological protein and further mediation of oxidative stress [52]. This is exacerbated by the dysregulation of furin transcription by iron. Furin is a serine protease which regulates α-secretase-mediated APP processing [245]. Furin mRNA levels have been reported to be downregulated in AD brains [245], and this has been attributed to excess iron accumulation. Iron-mediated downregulation of furin activity has therefore been proposed to increase amyloidogenic APP processing and Aβ plaque production [246]. Excessive iron has also been observed to lead to excessive tau phosphorylation deposition by upregulating GSK3β and ERK1/2 signalling pathways [247,248].

There is ample evidence suggesting that the levels of trace metals become dysregulated in AD, and numerous studies implicate these changes in the excessive accumulation of Aβ plaques, amyloidogenic processing and Aβ-mediated oxidative stress and ROS generation. Although not yet studied to as great an extent, metal ion dysregulation has also been proposed to exacerbate tau hyperphosphorylation and synaptic deficits, reflecting the multitude of pathological effects which arise as a result of metal dyshomeostasis.

## 3. Targeting Neuromodulator Function for AD Treatment

Current treatments for Alzheimer’s disease, aside from the controversially approved aducanumab, target aberrant neurotransmission which arises late in disease presentation, downstream of Aβ and tau aggregation, synaptic dysfunction, oxidative stress and metabolic and neuronal dyshomeostasis. Approved treatments consequently do little to reverse pathology, acting only to slow the inevitable progression of the disease. Furthermore, conventional drugs target just one component of this complex, multifactorial disorder, and do not show clinical effects in all patients due to pathogenic complexity and variability between cases. This makes choices regarding the initiation of treatment difficult, and there remains no official criteria on which decisions regarding drug administration can be based. Additionally, the majority of current treatments are cholinesterase inhibitors, which are known to present with unpleasant side effects such as nausea, gastrointestinal upset, muscle weakness and weight loss [13]. For these reasons, there is an urgent need to expand the focus of AD drug development. Neuromodulators have proven roles in the control of neuronal activity and synaptic plasticity. Critically, in addition to arising as a result of Aβ and tau aggregation, their dysfunction has been implicated as a key driver of disease progression. Many studies have suggested that aberrant neuromodulator function occurs upstream to alterations to neurotransmission and pathological protein accumulation. Therefore, there is growing curiosity surrounding the therapeutic potential of neuromodulators for AD. A summary of neuromodulator-targeting drugs discussed can be found in Table 1.

### 3.1. Neuropeptide Targeting

A vast array of neuropeptides display altered levels and function over the course of AD, leading to alterations in neuroprotective effects and synaptic plasticity, increased pathological protein accumulation and consequently cognitive disturbances relating to learning and memory, the sleep/wake cycle and feeding behaviours. For this reason, attempting to target and restore neuropeptide function is of high therapeutic interest. Due to a lack of reuptake mechanisms, neuropeptides tend to exert longer-lasting effects on relevant targets from large distances. This is difficult to achieve with neurotransmitter targeting, which is predominantly faster acting with rapid reuptake [89]. Neuropeptide targeting may therefore enable effects to reach a more substantial target area, without the requirement for frequent dosing. Furthermore, studies have proposed that alterations to neuropeptide function occur prior to the appearance of pathological Aβ and tau tangles [108,113,305], indicating that they may be one of the earliest contributors to disease progression. In this case, their targeting could be more beneficial than current treatments which tend to be prescribed later in disease stages.

Neuropeptide Y has been implicated in a plethora of processes associated with neuronal function in AD, including neuroprotection against excitotoxicity, Aβ plaque accumulation, synaptic plasticity alterations and changes to the excitatory/inhibitory balance. Consequently, a number of studies have looked to investigate its therapeutic potential in the treatment of the disease. Intracerebroventricular NPY injection in mice has been shown to protect against Aβ-induced deficits in spatial memory via the prevention of oxidative stress [249], whilst reports have shown that NPY administration reduces damaging excitotoxicity [250], inflammation [251] and neurodegeneration [252]. Neuropeptide Y targeting in AD is therefore a topic generating large levels of interest, and has already been discussed in a number of recently published review papers [305,306,307]. Targeting calcium channels present on NPY-expressing neurons to restore their function [113], or signalling pathways downstream of NPY action, such as the JNK signalling pathway [253,254,308] have also been investigated, and show therapeutic potential in the treatment of AD.

PACAP has been observed to induce non-amyloidogenic α-secretase-dependent processing of the amyloid precursor protein [255] and long-term daily nasal administration of the neuropeptide in APP(V717I) transgenic mice led to improved cognitive function and reduced presentation of pathogenic Aβ [256]. Similarly, VIP administration into 5xFAD mice as young as 1-month was able to protect against Aβ accumulation in cortical and hippocampal regions, and reduce brain atrophy [257], highlighting the potential benefits of neuropeptide targeting early in disease presentation. Oxytocin signalling has been associated with LTP and synaptic plasticity underlying learning and memory, and its intranasal administration in AD mice has been shown to improve performance on the Morris water maze, which was accompanied by significant reductions in Aβ and tau deposition as a result of reduced ERK1/2, GSK3β and AChE activity, and attenuated neuronal death [258]. A separate study has shown that oxytocin administration and reductions in ERK and nuclear factor kappa B (NKκB) signalling pathways in young APP/PS1 mice reduced inflammatory cytokine release and delayed hippocampal atrophy [259]. Additionally, administration of oxytocin has been shown to lead to its extensive distribution in the brain, enabling the targeting of multiple brain regions [309]. These studies highlight the potential of neuropeptides to treat multiple, widespread components of AD pathology, early in disease presentation. Various other neuropeptides, including insulin [260,261,310], CGRP [262,311], bradykinin [263] and opioid peptides [111] have been studied as useful therapeutic targets for AD, due to their reported roles in synaptic plasticity and neuronal function. Orexin antagonists have been suggested to restore AD-associated sleep disturbances in 5xFAD mice [264], although their effect on other cognitive processes, such as learning and memory, has yet to be explored.

The potential of peptide therapeutics in AD treatment is further amplified by recent studies which have aimed to enhance efficacy of peptide drug delivery using novel modifications and delivery strategies. For example, increased stability and half-life of circulating neuropeptide-based treatments have been achieved via N-terminal acetylation [312], N-methylation [313] and thioamidation [314]. The potential to apply these strategies to peptide-based therapeutics in an AD context further increases treatment possibilities and potential benefits. Additionally, overcoming blood-brain-barrier impermeability, a notorious limitation of drugs targeting the brain, has been accomplished via the conjugation of neuropeptide Y with apolipoprotein B. This resulted in widespread NPY expression in the central nervous system of APP transgenic mice, and reversed hippocampal neurodegeneration and associated learning and memory deficits [252]. Neuropeptide therapeutics therefore present vast opportunities for more effective management of AD progression, particularly during early disease stages.

### 3.2. Hormone Targeting

Alongside more recognised peripheral roles, glucocorticoids and sex hormones have been observed to underlie the control of neuronal activity and synaptic plasticity in the brain. Elevated GC levels have been correlated with AD presentation, leading to interest surrounding their targeting in treatment of the disease. Preventing excessive release of glucocorticoids may enable a critical upstream block on Aβ accumulation by repressing binding to the glucocorticoid responsive element in the promoter region of the β-secretase gene, thus preventing expression of the enzyme and promoting non-amyloidogenic APP processing. Glucocorticoid receptor (GR) antagonism via intraperitoneal mifepristone administration was shown to reduce β-secretase and pathological Aβ levels in the hippocampus of APP/PS1 mice after just three days of treatment, and this correlated with improved learning and memory [140]. Encouragingly, mifepristone treatment was able to exert these effects in mice as old as 12 months, indicating that reducing GC levels may still be capable of reversing pathology late in disease presentation, which has been difficult to achieve with conventional neurotransmitter-targeting treatments. Application of the drug was also shown to overcome synaptic deficits and episodic memory loss in young 4-month Tg2576 mice [265], indicating that treatment may be beneficial when applied during both early and late stages of AD. A separate study revealed that mifepristone application was able to reverse synaptic deficits in the hippocampus of 3xTg mice [266], whilst its application has also been shown to reduce tau pathology [267], highlighting the ability of the drug to target multiple components of pathology. One limitation of mifepristone however, is that it is a non-selective GR antagonist and consequently exerts off-target actions; binding of the drug to the progesterone receptor has been reported to result in undesirable side effects [80,266,315]. Encouragingly, the selective GR modulators CORT108297 and CORT113176 were also able to reverse pathological Aβ production, neuroinflammation and hippocampal atrophy, and rescue synaptic deficits by re-establishing the levels of synaptic proteins synaptotagmin and PSD95 in an acute mouse model of AD [266]. Therefore, selective GR antagonism may be a more beneficial way of reducing GC-mediated progression of AD pathology. The application of these molecules in chronic models of the disease, which better depict its pathological progression, would be interesting to observe.

The role of oestrogen in AD pathology remains widely debated amongst researchers, and further studies are required to elucidate how the effectiveness of hormone replacement therapy is impacted by other factors such as dose, time point, or individual risk factors. In their systematic review of hormone replacement therapy (HRT) for AD, Ibrahim et al., [316] concluded that commencement of HRT only prior to the onset of the menopause results in beneficial long-term effects for AD treatment. This corroborates with results from the Women’s Health Initiative Memory Study (WHIMS), which found that oestrogen therapy in post-menopausal women increased incidence of probable dementia [317]. Similarly, a ‘window of opportunity’ hypothesis surrounding oestrogen treatment for AD has been presented, as 3xTg mice were observed to display reduced Aβ burden and improved behavioural performance following administration at middle- (7–9 months), but not old-age (16–17 months) [268]. Previous reports have proposed that oestrogen has the potential to exert neuroprotective functions by reducing brain atrophy in prefrontal and medial temporal lobe regions in human patients [270]. Oestrogen administration has also been linked with the improved transmission of serotonin [271] and acetylcholine [272], as well as enhanced LTP magnitude correlating with improved cognitive performance of AD rats [269]. Application of the hormone or oestrogen receptor modulators has been proposed to reduce pathological Aβ accumulation [268,274,275,318] and promote tau clearance [273]. The therapeutic potential of oestrogen therapy in targeting multiple pathological elements of AD is unsurprising considering the reported neuroprotective role of the hormone. That said, it is clear that the timing of HRT administration must be well controlled in order to result in optimal therapeutic benefits, and further research is required to fully elucidate how effectiveness of HRT may be impacted by other factors.

Reducing levels of gonadotropins FSH and LH, which have been shown to positively correlate with disease progression and cognitive decline in AD, has also been suggested to display therapeutic potential in treatment of the disorder [87]. A recent study has shown that inhibiting the actions of FSH leads to blockage of the C/EBPβ–AEP/δ-secretase pathway [86]. This pathway has previously been implicated in the amyloidogenic processing of APP, resulting in the production of pathological Aβ aggregates [319]. The δ-secretase arginine endopeptidase (AEP) has also been shown to cleave tau, leading to neurofibrillary tangle production [320]. Blocking FSH ameliorated both Aβ and tau pathology in 3xTg mice. Increases in dendritic spine and synapse number, and a reduction in neuronal death by apoptosis were also observed, and these changes correlated with rescued spatial memory performance of mice on the Morris water maze [86]. Genetic ablation of the LH receptor (LH-R) has been shown to reduce Aβ plaque content in the hippocampus and cortex of APPswe mice [321]. This study also noted impacts of LH-R ablation on α7-nicotinic acetylcholine receptor expression, changes to which have been linked to Aβ accumulation-induced alterations to cholinergic transmission [322]. A separate study in 3xTg mice has shown that serum LH downregulation inhibited GSK3β signalling and increased BDNF transcription, resulting in rescued spatial memory [276].

Many studies have investigated the beneficial effects of testosterone administration in reducing AD pathology; T administration in mice previously injected with Aβ leads to improved learning and memory [280]. Underlying mechanisms for this neuroprotective effect have recently been proposed; Yan et al., [152] observed that T administration increases hippocampal pyramidal cell, dendritic spine and PSD95 levels, all of which are essential components of synaptic plasticity. These changes were reversed following androgen receptor blockade. Testosterone has also been observed to exert presynaptic neuroprotective effects from Aβ accumulation, by restoring presynaptic protein levels critical for synaptic transmission [74]. In terms of effects on pathological protein accumulation, testosterone has been shown to increase Aβ clearance by microglia, suppress Aβ-induced release of pro-inflammatory cytokines TNF-α and interleukin 1β (IL-1β) [277], and attenuate tau hyperphosphorylation via GSK3β inhibition [278]. Finally, T administration has been observed to preserve mitochondrial function and mitochondrial-ATPsynthase coupling following brain insult [279]. Therefore, a multitude of studies have highlighted the potential benefits of testosterone administration in restoring synaptic plasticity, and reducing pathological protein accumulation and neurodegeneration.

### 3.3. Neurotrophin Targeting

Due to its known role in synaptic plasticity, BDNF application is viewed as a plausible way to potentially overcome cognitive deficits associated with underlying alterations to synaptic remodelling and neuron growth [162]. Studies have shown that BDNF administration does indeed exert protective effects over both amyloid beta- and tau-associated neurotoxicity. Gene delivery of human BDNF carried in an adeno-associated virus (AAV) into P301L mice led to reduced neuron loss and alleviated synaptic degeneration, which correlated with the rescue of behavioural deficits associated with working memory [62]. Interestingly, the same study noted that BDNF application did not affect tau hyperphosphorylation levels, indicating that directly targeting AD pathological proteins themselves may not be necessary in order to overcome their related pathological effects on neuronal survival and plasticity. BDNF application has been suggested to stimulate non-amyloidogenic APP processing via the promotion of α-secretase processing of APP, as opposed to β-secretase cleavage [281], which is heavily implicated in the production of pathological, amyloidogenic Aβ [323]. Critically, neurotrophin targeting with small molecule mimetics may help to overcome drug limitations associated with BBB impermeability and the presentation of adverse effects [324]. For example, the application of the BBB-permeable small-molecule peptide mimetic Peptide 021 (P021) in 3xTg mice increased BDNF signalling and inhibited GSK3β-mediated tau hyperphosphorylation, which led to the rescue of synaptic deficits and associated cognitive decline [282].

Small molecule treatments which increase the initiation of TrkB and TrkC receptor signalling have also shown promising therapeutic signs. The application of TrkB/C-R ligands, LM22B-10 and PTX-BD10–2, has recently been reported to promote survival and outgrowth of cultured hippocampal neurons, and reduce tau pathology and cholinergic neuron degeneration in human-induced pluripotent stem cells following amyloid beta application [283,284]. Studies have investigated the benefits of targeting signalling molecules downstream of BDNF/TrkB-R activity; inhibiting the excessive activation of GSK3β and ERK2 using small molecule treatments has been shown to ameliorate tau hyperphosphorylation, neurodegeneration, synaptic deficits, astrogliosis and microgliosis in rodent models of AD [69,325]. Vitally, the application of these small molecules, which include geniposide [285], tolfenamic acid [286], isoorientin [70] and dimethyl fumarate [287] is also associated with improved cognitive function and memory. The administration of therapeutics targeting neurotrophin signalling, particularly that associated with BDNF/TrkB-R activation, is therefore becoming increasingly considered, and may be a more appropriate and promising way of treating AD pathology and neurotransmitter dysfunction upstream of their presentation. Crucially, the potential of BDNF-related targeting with small-molecule drugs may help overcome major limitations associated with drug delivery to the CNS, including off-target and associated side effects and BBB impermeability.

Despite a primary focus on the effects of BDNF treatment for AD, a handful of studies have suggested that targeting other neurotrophins may also be useful. The intracerebroventricular infusion of NGF, NT-3 and NT-4 was shown to reverse spatial memory deficits of aged rats which, in the case of NGF and NT-3, correlated with significant reductions in cholinergic atrophy in the basal forebrain [288]. The same study noted that increased cholinergic neuron size was observed in the hippocampal-projecting medial septum and vertical limb of the diagonal band of Broca, implying that neurotrophins may enhance cognition by strengthening cholinergic-dependent hippocampal synaptic activity. The potential of NGF to protect cholinergic neurons from degeneration was investigated in a phase 1 study in AD patients via the stereotactic gene delivery of AAV2-NGF into the nucleus basalis, a region of the basal forebrain that has widespread projections to hippocampal and cortical regions and has been heavily implicated in the cholinergic hypothesis of AD [326]. Results indicated that the treatment was well-tolerated and halted accelerated decline of performance on neuropsychological tests [289], emphasising the potential of neurotrophin targeting for AD. The above studies highlight alternative ways of overcoming aberrant neurotransmission without targeting neurotransmitter systems, preventing off-target effects and enabling a wider range of pathology to be addressed by a single treatment avenue. The ways in which alternative neurotrophins may be harnessed using small-molecule mimetics in order to overcome BBB impermeability is relatively understudied compared to that of BDNF, so further research into this area may present new opportunities for AD therapeutics.

### 3.4. ATP Targeting

Alterations to brain metabolic activity and ATP production have been suggested to occur prior to the production of Aβ and tau aggregates in AD, making these processes upstream targets for the prevention and reduction of disease pathology. Enhancing intracellular levels of ATP would enable the restoration of ATPase activity and balance of ion concentrations which underlie neuronal activity, and would therefore prevent excitotoxicity associated with excessive calcium influx into the neuronal membrane. An increase in intracellular ATP concentration has been achieved via the application of calcium polyphosphate microparticles, or Na+ salt of polyphosphate complexed to calcium, which are BBB-permeable and physiologically occurring. This led to protection of neurons against impaired energy homeostasis and Aβ-induced cell death [290]. Increasing ATP production can also be attempted by the targeting of F_1_F_O_-ATPase, which displays altered function in AD resulting in limited ATP generation in the mitochondrial inner membrane. Two substances, J147 and Salvianolic acid B (SalB), have so far been shown to effectively modulate F_1_F_O_-ATPsynthase to increase ATP production in an AD context [291,293,327]. J147 is currently undergoing clinical trials [327], after it was previously shown to improve memory in aged AD mice by increasing BDNF and NGF levels [292], revealing therapeutic effects in addition to actions on ATP generation. SalB has been observed to exert neuroprotective effects by increasing Aβ clearance [294,295,296]. Currently, further research is required to fully elucidate the mechanisms of action of both drugs, but they show early promise in enabling the increased production of intracellular ATP to treat AD.

Mitochondrial dysfunction is a critical contributor to suboptimal ATP production, so strategies which aim to restore mitochondrial function are likely to be of therapeutic benefit. A recent study has proposed that upregulation of the mitophagy-associated protein; PTEN-induced putative kinase 1 (PINK-1), is able to restore mitochondrial function and attenuate Aβ accumulation in the mAPP mouse model by increasing clearance of Aβ through the promotion of lysosome recruitment and autophagy signalling [203]. The same study confirmed that increasing PINK-1 levels correlated with improved learning and memory of mAPP mice. PINK-1 levels are significantly reduced in the hippocampus of post-mortem brain samples from AD patients, confirming the therapeutic potential of PINK-1 targeting in restoring mitochondrial function and ATP production to optimise Aβ clearance. A major contributor to deficits in ATP production is the metabolic switch from oxidative phosphorylation to glycolysis in neuronal and microglial mitochondria. Studies have looked to reverse this switch, which appears dependent on increased PKM2 levels, in order to augment ATP production. Pan et al., [54] pharmacologically inhibited PKM2 using shikonin or compound 3K, both of which led to the suppression of glycolytic gene expression in hippocampal microglia of 5xFAD mice, and the upregulation of PKM1 activity and oxidative phosphorylation. Microglial activation was reduced, pro-inflammatory cytokine release was attenuated, and these effects correlated with decreased Aβ plaque burden and improved cognitive performance.

Downstream of reduced ATP generation, the re-establishment of ion concentration gradients which become altered due to reductions in ATP production, oxidative stress and Aβ binding to transmembrane P-ATPases may help to prevent excitotoxic effects arising from excessive calcium influx into neurons [328]. However, due to the essential roles of P-ATPases in neuronal function, their direct manipulation or blockage to prevent calcium influx would be risky. One study has proposed that the use of small-molecule mimetics to prevent Aβ oligomer interaction with the Na/K-ATPase α3-subunit by covering the Aβ interacting surface may be a more beneficial way of rescuing Na/K-ATPase function [329]. The effectiveness of this interesting strategy has yet to be fully explored.

Overactivation of P2X receptors by excess extracellular ATP has been shown to contribute to excitotoxicity and the promotion of neuroinflammation in AD. Blockade of these receptors is therefore of therapeutic relevance in treatment of the disease, particularly that of the P2X7 receptor which is a key modulator of neuroinflammatory processes [330]. Antagonism selective for P2X7 receptors is generally difficult to achieve, but one compound, the Brilliant Blue G (BBG) antagonist, has been reported to display 1000-fold increased potency at P2X7 receptors than other P2X receptor subtypes [331] and has thus been tested in an AD context. Rats previously injected with Aβ displayed rescued memory impairment following intraperitoneal BBG administration, and application of the antagonist to cell culture reversed Aβ-mediated dendritic spine loss [297]. BBG application in J20 mice attenuated Aβ plaque presentation in hippocampal regions due to inhibition of GSK3β and subsequent upregulation of α-secretase and non-amyloidogenic APP processing [298]. Additionally, BBG has been reported to display high BBB permeability [332], furthering the therapeutic potential of the molecule in AD treatment. The impact of P2X receptor antagonism in human AD patients, and the effects of this blockade on other components of pathology, such as synaptic loss and tau hyperphosphorylation, have yet to be determined.

### 3.5. Metal Ion Targeting

As the role of transition metals is becoming increasingly implicated as a key contributor to neurodegeneration in Alzheimer’s disease, more research is being focused on finding drugs that could block their harmful effects. The introduction of metal chelation in the hope of reducing the irregular accumulation of metals is termed as chelation therapy [333]. Metal chelating compounds have gained interest for their roles in binding and removing excess metal ions via urination and fecal excretion.

Recently, a rhodamine-based metal chelator which captures copper ions was shown to prevent the formation of toxic amyloid aggregates in the brain and inhibit metal-induced ROS production [299]. This gives rise to the possibility of the inhibition of metal-mediated toxicities in the treatment of Alzheimer’s disease. Having previously been shown to reduce Aβ deposition in rodent models and neuronal cell cultures [301,302], iodochlorhydroxyquin (clioquinol), a metal chelating agent that has affinity for copper and zinc and is able to cross the blood brain barrier, has been observed to have positive effects in AD patients by slowing their cognitive decline and plasma Aβ levels [300]. Similarly, the administration of DP-109, which has affinity for copper and zinc ions, resulted in reduced amyloid beta plaque burden in the brains of transgenic AD mice [303].

There are studies on brain-permeable iron chelators, for example VK-28, that show potential in exerting neuroprotective effects in rodent models [304]. It was discovered that the treatment of synthetic amyloid-beta fibrils with iron chelator deferoxamine (DFO) significantly decreased Aβ-associated neurotoxicity [49]. Furthermore, DFO was found to suppresses tau hyperphosphorylation by reducing iron-mediated activation of GSK3β in APP/PS1 mice [247]. Moreover, clinical trials have demonstrated the effectiveness of deferoxamine in slowing AD progression in human patients [334]. However, due to toxicity and restricted permeability across the BBB, the use of presently available iron chelators is limited; while the majority of small molecular weight lipophilic iron chelators are able to cross the blood brain barrier, they exhibit high toxicity. On the other hand, iron chelators with lower toxicity have lower lipophilicity, and are unable to cross the BBB as easily [335].

## 4. Boosting Efficacy and Reducing Limitations of AD Therapeutics

Several studies surrounding neuromodulator-associated therapeutics have aimed to further enhance drug efficacy by targeting multiple elements of disease progression via the use of multi-targeted directed ligands (MTDLs) and combinatorial drugs. MTDLs are single molecules with a plethora of targets, usually encompassing anticholinesterase activity plus additional protective properties such as metal chelation, anti-oxidant effects and prevention of excitotoxicity [336]. Examples relevant to AD include α-mangostin [337], a natural xanthone from mangosteen fruit, which has been shown to act as a cholinesterase inhibitor [338]. In addition, its application has been observed to improve mitochondrial function and reduce oxidative stress in a Parkinson’s disease in vitro context [339], and to exert copper-chelating properties [340] which may help attenuate metal-induced Aβ aggregation [301]. The molecule has also been reported to display anti-inflammatory effects [341]. The essential trace element selenofuranoside, as well as displaying cholinesterase inhibition [342], was shown to restore hippocampal Na/K-ATPase activity, correlating with improved long-term memory of rats [343]. A number of metal-chelating ligands with effects on additional components of AD pathology have been generated. These include compounds such as 8-hydroxyquinoline and beta-aminopyridine, which have been observed to restore oxidative stress associated with metal dyshomeostasis and free radical generation, as well as inhibit the formation of Aβ plaques via β-secretase inhibition and increase levels of acetylcholine via AChE inhibition (the potential of these molecules has been comprehensively reviewed by Santos et al., [344]). The conjugation of iron chelator DFO with 6-quinolinamine has been shown to display AChE inhibition as well as iron-chelating properties [345]. Finally, the synthesis of dimeric tacrine(10)-hupyridone (A10E), involving the linkage of tacrine with the naturally occurring alkaloid huperzine-A, resulted in an AChE-inhibiting tacrine derivative with additional BDNF activation properties. This compound was shown to prevent cognitive decline and Aβ oligomerization in APP/PS1 mice with higher effectiveness than either substance alone [346]. Critically, administration of the compound did not result in obvious hepatotoxic effects, which currently limit the lone use of tacrine to treat AD.

Combining neuromodulator-targeting with conventional neurotransmission-associated treatment options may reduce pathology and improve cognition to a greater extent than current treatment administration alone. The combined delivery of oxytocin and galantamine was observed to be more beneficial in suppressing Aβ and tau aggregation and restoring cognitive deficits in young AD rats than the application of just galantamine, due to increased inhibition of AChE, ERK1/2, GSK3β and caspase-3 [258]. This may occur as a result of activation of the PI3K/AKT pathway by oxytocin, which directly suppresses GSK3β activation and subsequent tau hyperphosphorylation [347]. The combined application of AChE inhibitor donepezil with cerebrolysin; a mixture of neurotrophins including BDNF and NGF purified from the pig brain, increased dendritic spine length and density in prefrontal and hippocampal regions of 2–9 month mice more so than respective treatments alone [348]. As previously discussed, oestrogen administration during a critical time window in early AD disease stages may be useful in preventing cognitive decline. However, administration of donepezil alongside the hormone has been shown to restore the ability of oestrogen to improve cognitive performance in very old as well as middle-aged rats [349]. This shows the potential of combinatorial treatments in reversing cognitive decline during more advanced stages of AD, which current drug treatments are unable to do. Combinatorial treatments for AD which harness neuromodulator function in combination with current neurotransmitter-altering drugs therefore show promise in increasing the potency of neuroprotective effects, and thus open multiple new avenues for improved AD therapeutics.

A further advancement surrounding AD therapeutics research is the use of novel nanoparticle encapsulation technology, a strategy notorious for its success in the recent rollout of the COVID-19 vaccine [350]. Due to the hydrophobic exterior of nanoparticles, this delivery technique overcomes common limitations associated with drug delivery to the CNS, most critically BBB impermeability. Additionally, nanoparticle delivery displays limited toxicity to neurons, and results in few adverse effects, another advantage over conventional treatments which are known to present with unpleasant side effects [13]. Consequently, improved targeted nanoparticle delivery of current AD drugs into the brain has already been moderately studied, and has been extensively reviewed by Fonseca-Santos et al., [351]. Regarding neuromodulator function, nanoparticle delivery of plasmid encoding BDNF to the brain has been shown to increase BDNF expression in older (6–9 month) APP/PS1 mice twofold, leading to a 40% reduction in Aβ accumulation [352]. Presynaptic and postsynaptic proteins synaptophysin and PSD95 were also observed to be upregulated by BDNF delivery, which may help restore synaptic plasticity changes associated with reductions of BDNF in the hippocampus. Huperzine-A, which is approved for AD treatment in China, has been observed to increase secretion of NGF and BDNF [353,354], and was shown to significantly improve cognitive function of memory-impaired scopolamine-treated mice when applied inside a nanocarrier-based gel to the skin [353]. Huperzine-A, which is well-tolerated with limited side effects, can also be considered an MTDL, considering its inhibitory effects on anticholinesterase activity [355]. Nanoparticle iron chelators have been presented as a potential therapeutic approach in overcoming metal dyshomeostasis and associated oxidative stress and neurodegeneration in AD [356]. Due to their permeability across the blood brain barrier, the use of nanoparticles allows non-lipophilic chelators with larger molecular weight to be delivered into the brain, while reducing any potential toxicities associated with lipophilic chelating agents [357]. More studies are required to fully elucidate the possibilities associated with nanoparticle drug delivery in AD, and how efficient they may potentially be at overcoming limitations of conventional drug delivery. This delivery method shows promise in the appliance of therapeutics which exert efficacious effects on AD pathology, but are unable to cross the blood-brain-barrier.

## 5. Conclusions

It is undeniable that targeting neuromodulator function in AD displays potential, reflected by the substantial volume of studies exhibiting effectiveness of neuromodulator-targeting drugs in rescuing AD pathology and synaptic plasticity deficits underlying cognitive dysfunction. This is unsurprising, considering the fundamental and extensively-reported roles of neuromodulators in neuronal activity and homeostasis. Increased research interest into this area brings light to a situation in which current treatments for AD, predominantly targeting neurotransmitter function, are well-known to be insufficient in doing more than simply slowing the inevitable spread of disease in the brain.

A critical benefit of targeting neuromodulators is that many of their actions occur upstream to pathological effects. Therefore, their targeting may enable a larger cascade of downstream effects to be impacted, encompassing several pathological components including synaptic plasticity changes, neurotransmission and Aβ and tau accumulation (Figure 1). This helps to overcome a major limitation of conventional AD drugs which target single components of the disease during later stages, rendering its prevention or reversal impossible. The effectiveness of therapies associated with neuromodulator function can be further advanced via their integration with more novel strategies, such as the development of multi-targeted directed ligands, combinatorial treatments and nanoparticle delivery, which aim to boost efficacy, limit the presentation of side effects and increase BBB permeability of conventional therapeutics. To date, the majority of neuromodulator-targeting treatments have only been studied in cultured cells or rodent models of AD (Table 1) and the potential of these drugs in human patients is yet to be determined. This highlights the plethora of therapeutics which may yet provide opportunity for improvement in the search for more effective Alzheimer’s disease treatments.

## Figures and Tables

**Figure 1 biomedicines-10-03064-f001:**
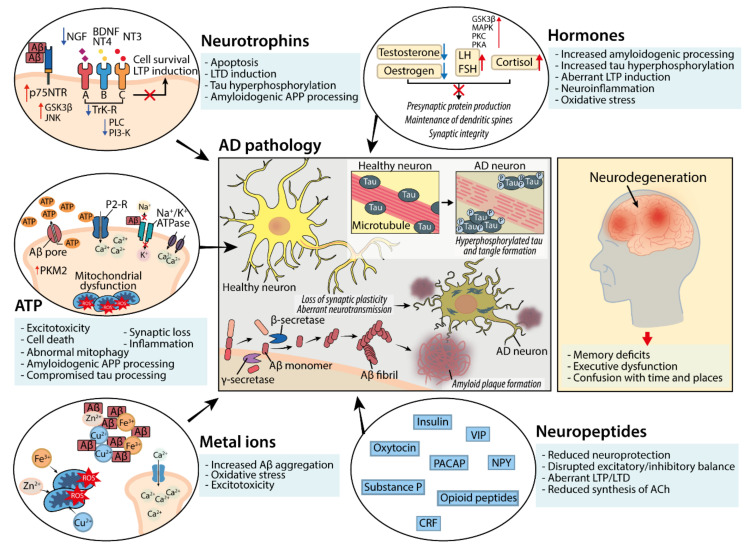
Summary of neuromodulator dysfunction in Alzheimer’s disease. Alterations to neuromodulator function are being increasingly regarded as critical drivers of AD pathology and resulting neurodegeneration and cognitive deficits. Alterations to levels of neuropeptides bring about compromised neuroprotection, reduced synthesis of neurotransmitters such as acetylcholine, disrupted control of excitatory and inhibitory network balances and aberrant LTP and LTD [26,27,28,29,30,31,32,33,34,35,36,37,38,39,40,41,42,43,44]. The accumulation of metal ions with Aβ exacerbates excitotoxicity and oxidative stress [45,46,47,48], and contributes to the production of toxic ROS [49,50,51,52]. Increased extracellular and reduced intracellular ATP as a result of metabolic switching [53,54] and Aβ-induced pore production in the neuronal membrane [55] underlie dysfunction of P-ATPases, [56], P2-R overactivation, excess calcium influx, mitochondrial dysfunction, abnormal mitophagy, excitotoxicity and cell death [57,58,59]. Reduced neurotrophin activation of TrK-Rs leads to increased apoptosis and LTD and reduced cell survival and LTP [60,61,62,63,64,65,66,67,68]. Activation of the p75NTR by Aβ activates GSK3β, which upregulates amyloidogenic APP processing and tau hyperphosphorylation [69,70,71,72]. Reduced levels of oestrogen and testosterone compromise synaptic integrity [73,74,75,76,77,78], whilst increased levels of FSH, LH and cortisol activate GSK3β, MAPK, PKC and PKA signalling, leading to amyloidogenic processing and tau hyperphosphorylation, oxidative stress and neuroinflammation [79,80,81,82,83,84,85,86,87].

**Table 1 biomedicines-10-03064-t001:** Potential neuromodulator-based therapeutics for Alzheimer’s disease and their reported impacts on pathology.

Neuromodulator Group	Name of Treatment	Stage Tested	Impact on AD Pathology
**Neuropeptide**	Neuropeptide Y	Rodent models and cultured cells	Protects against Aβ-induced spatial memory deficits, oxidative stress [249], excitotoxicity [250], inflammation [251] and neurodegeneration [252].
	Nimodipine	Rodent models	Restores NPY-expressing neuronal function and associated neuroprotection via targeting of Aβ-mediated Ca^2+^ channel disruption [113].
	SP600125	Rodent models	Reduces Aβ plaque burden, tau hyperphosphorylation, synaptic loss [253] memory impairment and apoptosis [254] via inhibition of JNK phosphorylation.
	PACAP	Rodent models and cultured cells	Promotes α-secretase-dependent APP processing [255], improves cognition and reduced pathogenic Aβ burden [256].
	VIP	Rodent models	Protects against Aβ accumulation, reduces brain atrophy [257].
	Oxytocin	Rodent models	Improves memory retention, reduces Aβ and tau deposition, ERK1/2, GSK3β and AChE activity and neuronal death [258]. Reduces inflammatory cytokine release and hippocampal atrophy [259].
	Insulin	Rodent models	Partially reduces levels of phosphorylated tau and improves learning ability [260]. Reduces pathological Aβ production and improves memory [261].
	BIBN 4096 BS (BIBN)	Rodent models	CGRP antagonist, increases expression of PSD95, reduces neuroinflammation, Aβ and tau pathology [262].
	des-Arg9-[Leu8]-bradykinin (DALBK)	Rodent models	Bradykinin 1 receptor antagonist, reverses spatial learning and memory deficits [263].
	UFP-512	Cultured cells	Specific agonist of δ-opioid receptors. Exerts neuroprotective actions by attenuating β-secretase expression [111].
	Dual orexin receptor antagonist (DORA-22)	Rodent models	Improves sleep [264].
**Hormone**	Mifepristone	Rodent models	Glucocorticoid receptor antagonist, reduces β-secretase expression and Aβ production, improves learning and memory [140]. Reverses synaptic deficits [265,266], reduces tau pathology [267].
	CORT108297 and CORT113176	Rodent models	Selective glucocorticoid receptor antagonists. Reverse Aβ production, neuroinflammation, hippocampal atrophy, synaptic deficits. Re-establish levels of synaptotagmin and PSD95 [266].
	Oestrogen (hormone replacement therapy)	Rodent models and human AD patients	Commencement during early stages/before the menopause reduces Aβ plaque burden, improves behavioural performance [268], enhances LTP [269], reduces brain atrophy [270], improves neurotransmission [271,272], promotes tau clearance [273].
	STX	Rodent models and cultured cells	Oestrogen receptor modulator, reduces Aβ levels, associated mitochondrial toxicity, and synaptic deficits and improves spatial memory [274,275].
	FSH-Ab	Rodent models	Reduces FSH levels, blocks δ-secretase-mediated amyloidogenic APP processing and tau neurofibrillary tangle production. Increases dendritic spine and synapse number. Improves learning and memory [86].
	Leuprolide acetate	Rodent models	Reduces LH levels, inhibits GSK3β signalling, increases BDNF transcription, rescues spatial memory [276].
	Testosterone	Rodent models and cultured cells	Increases dendritic spine number, PSD95 [152], presynaptic protein levels [74], Aβ clearance [277]. Reduces tau hyperphosphorylation via GSK3β inhibition [278], preserves mitochondrial function [279]. Improves memory retention [280].
**Neurotrophin**	BDNF	Rodent models and cultured cells	Reduces neuron loss and synaptic degeneration, rescues working memory deficits [62]. Stimulates non-amyloidogenic APP processing [281].
	Peptide 021	Rodent models	Increases BDNF signalling, inhibits GSK3β-mediated tau hyperphosphorylation, reduces synaptic deficits and associated cognitive decline [282].
	LM22B-10 and PTX-BD10-2	Rodent models and cultured cells	Activate TrKB and TrKC receptors. Promote survival and outgrowth of hippocampal neurons, reduce tau pathology and cholinergic neuron degeneration [283,284].
	Geniposide	Rodent models	Suppresses ERK signalling, reduces inflammatory cytokine release and augments synaptic plasticity [285].
	Tolfenamic acid	Rodent models	Inhibits GSK3β. Attenuates memory deficits, decreases tau hyperphosphorylation [286].
	Isoorientin	Rodent models	Selective GSK3β inhibitor. Attenuates tau hyperphosphorylation, Aβ deposition and neuroinflammation. Improves LTP and spatial memory [70].
	Dimethyl fumarate	Rodent models	Modulates GSK3β activity. Decreases tau phosphorylation and neuroinflammation, increases BDNF expression [287].
	NGF	Rodent models and human AD patients	Reverses spatial memory deficits, reduces cholinergic atrophy [288]. Halts accelerated decline of neuropsychological test performance [289].
	NT-3	Rodent models	Reverses spatial memory deficits, reduces cholinergic atrophy [288].
	NT-4	Rodent models	Reverses spatial memory deficits [288].
**ATP**	Inorganic polyphosphates	Rodent models and cultured cells	Protect against Aβ-mediated neurotoxic effects by enhancing intracellular ATP levels [290].
	J147	Rodent and human protein models	Modulates activity of ATP synthase [291], increases BDNF and NGF expression and learning and memory ability [292].
	Salvianolic acid B	Rodent models and cultured cells	Rescues Aβ-mediated dysfunction of ATP synthase and mitochondrial stress [293], decreases Aβ accumulation and neuroinflammation and improves cognitive function [294,295,296].
	PTEN-induced putative kinase 1 (PINK-1)	Rodent models	Mitophagy-associated protein, restores mitochondrial function and attenuates Aβ accumulation. Improves learning and memory [203].
	Shikonin and compound 3K	Rodent models	PKM2 inhibitors. Reduce microglial glycolysis and pro-inflammatory activity and Aβ plaque content [54].
	Brilliant Blue G (BBG)	Rodent models and cultured cells	P2X7 receptor antagonist. Rescues dendritic spine loss and memory impairments [297]. Attenuates Aβ plaque production via inhibition of GSK3β and activation of α-secretase [298].
**Metal ion**	Rhodamine-B-based compound (Rh-BT)	Rodent models and cultured cells	Copper chelator. Prevents formation of toxic amyloid aggregates and inhibits metal-induced ROS production [299].
	Iodochlorhydroxyquin (clioquinol)	Rodent models, cultured cells and human AD patients	Copper/zinc chelator. Slows cognitive decline and plasma Aβ accumulation in AD patients [300]. Reduces deposition of Aβ [301,302].
	DP-109	Rodent models	Zinc/copper chelator. Reduces amyloid beta plaque burden in the brain [303].
	VK-28	Rodent models	Iron chelator. Exerts neuroprotective effects to reduce neurodegeneration [304].
	Deferoxamine (DFO)	Rodent models and cultured cells	Iron chelator. Decreases Aβ-associated neurotoxicity [49] and suppresses tau hyperphosphorylation by reducing activation of GSK-3β [247].

## Data Availability

Not applicable.

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
