# Peer review of "Alternative Pharmacological Strategies for the Treatment of Alzheimer’s Disease: Focus on Neuromodulator Function"

_biomedicines, 2022, doi:10.3390/biomedicines10123064_

Round 1
Reviewer 1 Report
This paper will interest the large class of neuroscientists who perform neuromodulator functions in Alzheimer's disease treatment. It provides a helpful summary of alternative therapies for Alzheimer’s disease. In this manuscript, the author summarizes five different neuromodulators (neuropeptides, neurotrophins, hormones, ATP, and metal ions) functions in the central nervous system and dysfunction in Alzheimer's disease. As we all know, regular AD drugs have unpleasant side effects. They also state that the neuromodulator targeting medications for Alzheimer’s disease. The references well support the critical claims of the manuscript. However, a minor concern should be addressed.
Page 13, line 1: The health and AD neurons were not apparent to the readers in Figure 1. I recommend the author add “Health neuron” in the bright yellow neuron and “AD neuron” in the dark green neuron.
Reviewer 2 Report
Review of a manuscript “Alternative pharmacological strategies for the treatment of Alzheimer’s disease: focus on neuromodulator function” by Grace Cunliffe and coauthors submitted to “Biomedicines”
Alzheimer’s disease is the most prevalent neurodegenerative disorder for which there is no disease-modifying treatment. Current attempts to cure this disorder are insufficient at reversing pathology and are able only to slow the unavoidable advance of the disease. Due to high frequency of Alzheimer’s disease and lack of efficient treatment new approaches to cure this disorder should be developed. The authors review recent data on neuromodulators in patients with Alzheimer’s disease, combine the results on the changes of neuromodulation in patients and consider how novel therapies may be developed acting upstream to classical neurotransmitter systems. This is an important review of recent publications which will be interesting for the readers of “Biomedicines”. The following corrections and additions should be done.
Abstract
This following two long sentences should be split into two for easier reading:
Lines 21-22: “Neuromodulators are substances released by neurons which influence neurotransmitter release and signal transmission across synapses, and include neuropeptides, hormones, neurotrophins, ATP and metal ions.”
Lines 25-28:” Combining neuromodulator targeting with more novel methods of drug delivery, such as the use of multi-targeted directed ligands, combinatorial drugs and encapsulated nanoparticle delivery systems may help to overcome limitations of conventional treatments, which include difficulty crossing the blood-brain-barrier and the exertion of effects on a single target only.”
Introduction
Lines 44-45: ”Despite such concerning figures, due to its complex and multifactorial nature, the exact causes of AD are yet to be elucidated, and current treatments remain insufficient at reversing pathology and act only to slow the inevitable progression of the disease and its associated symptoms.”
After this sentence the authors should add the following text and a reference: “Patients with Alzheimer’s disease often have comorbidity with other diseases making the development of treatment a difficult challenge” [reference: A New Link Between Diabetes and AD. Cell Mol Neurobiol. 2020 Oct;40(7):1059-1066. doi: 10.1007/s10571-020-00796-4. PMID: 31974905].
Lines 31-32: ”As a result, the drug was rejected for use by the European Medicines Agency [23], and remains yet to be approved outside of the United States.”. The sense of this sentence is blur; it should be better shaped and rewritten more clearly.
Lines 52-53:” Neuromodulator signalling therefore impacts widespread and numerous components of AD disease progression” should be corrected as follows:” Neuromodulator signaling therefore impacts widespread and numerous components of AD progression”.
Lines 28-31: ”For example, orexin has been shown to couple to Gq protein transducers, which results in the activation of phospholipase C, leading to increased cytosolic calcium influx, as well as the downstream activation of protein kinase C and the extracellular signal-regulated kinase (ERK) 1/2 signalling pathway [27], which has been strongly implicated in the formation and retrieval of long-term memory [28].”
This very long sentence is hard to read. It should be split into at least two.
2.7. ATP Lines 35-53. This section is overloaded by the very basic information concerning ATP and mitochondria.
It should be truncated and presented in a more concise and succinct way. Section 2.8 is more relevant to the topic of the review.
2.9. Metal ions. Similar corrections are needed for the section 2.9. which should be presented in a more succinct manner, since more related to the topic information is in the section 2.10. Metal ions; dysfunction in AD.
Conclusion.
Line 32. A critical benefit of targeting neuromodulators is that many of their actions occur upstream to pathological effects, instead of as a consequence of them.
This sentence is not very clear (instead of as a consequence of them) and needs clarification.
Overall a very interesting review which needs some corrections and more concise presentation in several place.
